# Multi-Technology Multi-Operator Site Sharing: Compliance Distance Analysis for EMF Exposure

**DOI:** 10.3390/s23031588

**Published:** 2023-02-01

**Authors:** Mohammed S. Elbasheir, Rashid A. Saeed, Salaheldin Edam

**Affiliations:** 1School of Electronic Engineering, College of Engineering, Sudan University of Science and Technology, Khartoum 14413, Sudan; 2Department of Computer Engineering, College of Computers and Information Technology, Taif University, P.O. Box 11099, Taif 21944, Saudi Arabia

**Keywords:** EMF exposure, power density, total exposure ratio, compliance boundary, 5G, massive MIMO

## Abstract

In recent years, the development of wireless technologies has led to fast growth in mobile networks, especially with the rise of 5G New Radio (5G NR). A huge number of base stations (BSs) are mandatory to serve the growth of mobile services, which has led to concerns about the increase in electromagnetic field (EMF) radiation exposure levels. To control the overall power emitted by EMF transmitters, international bodies have set maximum exposure limits. This paper investigates the compliance distances (CDs) of shared sites by a group of Mobile Network Operators (MNO) as multi-operators operating with multi-technology and sharing the same tower. The study investigated the CDs of the most two commonly used types of sharing sites, macro and indoor-Based solution sites (IBS). In addition, the study analyzed the power densities and total exposure ratios for the general public and occupational workers in each sharing scenario. The results showed that, compared with a single MNO, the CD increased by 41% in the case of two MNOs, 73% for three MNOs, and 100% for four MNOs. The EMF site sharing scale-up formula was used to estimate the increase in CDs for N number of MNOs assuming that all MNOs use the same site configuration. In addition, the results showed that 5G has the highest contribution to the total exposure ratio (TER) at the CD in the main direction of the antennae.

## 1. Introduction

Mobile networks are the fastest-growing wireless communication networks due to the massive increase in data service consumption, especially with the wide spread of smartphones and tablets. This leads to the generation of large data traffic over the mobile network that will reach very high figures by 2030 compared with 2020 [1,2]. Expanding mobile networks necessitates the installation of additional BSs to increase network coverage and capacity. Most deployed sites involve the 2G Global System for Mobile Communication (GSM), 3G Universal Mobile Telecommunications System (UMTS), 4G Long-Term Evolution (LTE), and 5G NR, which has recently been introduced. The installation of more BSs raises electromagnetic field exposure levels, which has become a source of concern due to the effects on human health [3,4,5]. Several well-known international organizations have set standard guidelines, i.e., the International Commission on Non-Ionizing Radiation Protection (ICNIRP) [6,7] and the Federal Communication Commission (FCC) in the USA [8]. These standards have been adopted by national regulators in many countries to control the installation of EMF transmitters. The ICNIRP and FCC guidelines distinguish between the general public and occupationally exposed individuals. The general public comprises normal individuals who are exposed to EMFs, and occupational personnel are the staff exposed to certain situations associated with EMFs and are well-trained for possible risks [6].

It is predicted that 5G technology will be the system for general purposes [9] because 5G provides a high capacity and enables many features [10]. This requires the installation of NR BSs at higher-frequency bands [11], and most are colocated with existing technologies (2G, 3G, and 4G). The group of solutions and technologies operating at one site causes an increase in EMF exposure, and investigation is needed to evaluate the total accumulated radiation and assess it versus the standard limits [12,13,14].

In many countries, the MNOs have started to share mobile sites to expand their network services in terms of coverage footprint and capacity resources [15]. Site sharing has become a good alternative and a key approach because of the following main reasons [16]:It is less expensive to share a site than to build a new one. According to [17], infrastructure sharing can result in significant cost savings—up to 40% of assets can be saved by site sharing, and cash flow can improve by 31% as a result.The densification of the existing sites increases the difficulty of acquiring more physical sites within the required nominal locations and leads to fewer options.Site sharing enables the rationalization of the legacy 2G and 3G networks, taking into account the declining revenues from 2G and 3G networks and the higher spectral efficiency of the next generation of 4G and 5G technologies.Site sharing enables diverting the investment to other important innovations, such as the deployment of more 5G sites.Site sharing has some social benefits because it lowers network costs, which lowers the customer’s service fee.Tower sharing benefits the environment by reducing the number of sites with better looks and views.

Network sharing can proceed in many ways, starting from a small part of the infrastructure up to full sharing, including core parts [18]. Sharing can be classified into two groups. The first is passive sharing, which mainly involves sharing parts of the tower site, antennas, and/or the backhaul part (MW, transport, fiber, etc.), as illustrated in Figure 1.

The second group is active sharing, which consists primarily of sharing the Radio Access Network (RAN) component with a dedicated frequency spectrum for each operator as the Multi-Operator RAN (MORAN) and a shared frequency spectrum as the Multi-Operator Core Network (MOCN), as shown in Figure 2.

Passive site sharing raises concerns about the total EMF exposure radiated from the antennae because each operator has separate radio equipment and all of them transmit from the same location. In this study, we determined the total accumulated exposure by calculating and analyzing the CDs for the two types of site sharing at the macro site and the indoor IBS site.

The merit of this analysis is in the calculations that consider the real operating setup (typical configurations) and involve the actual realistic transmitted power. Additionally, the goal was not to compare the levels of exposure to different technologies. Instead, it was to find out what the CDs are when different technologies and operators all transmit in the same place.

This manuscript is structurally organized into three main sections: the exposure standard limits, CD calculations, and the results and discussion for site sharing at macro and indoor sites.

## 2. Exposure Standard Limits

The guidelines of the ICNIRP are the most widely adopted and used for controlling non-ionizing radiation exposure in many countries [19,20,21]. This standard has set maximum limits based on baseline values directly related to the adverse effects produced by EMF radiation. The ICNIRP defined two groups of people for radiation exposure [7], occupational workers (OWs) and the general public (GP). Similarly, the USA FCC [8] defined the maximum permitted exposure (MPE) for the general public and occupational workers. Table 1 lists the exposure limits for both the ICNIRP and FCC.

The ICNIRP also specified the total reference limits from instantaneous concurrent exposures from multiple sources, which are aggregated using Equation (1) shown below [6]:(1)$$\begin{array}{c} \overset{30\;\mathrm{MHz}}{\underset{i=100\;\mathrm{kHz}}{\sum }}\left\{{\left(\frac{E_{inc,f_{i}}}{E_{\mathrm{inc},\mathrm{RL},f_{i}}}\right)}^{2}+{\left(\frac{H_{\mathrm{inc},f_{i}}}{H_{\mathrm{inc},\mathrm{RL},f_{i}}}\right)}^{2}\right\}\\ \;+\overset{2\;\mathrm{GHz}}{\underset{i>30\;\mathrm{MHz}}{\sum }}\mathrm{MAX}\left\{{\left(\frac{E_{inc,f_{i}}}{E_{\mathrm{inc},\mathrm{RL},f}{}_{i}}\right)}^{2},{\left(\frac{H_{\mathrm{inc},f_{i}}}{H_{\mathrm{inc},\mathrm{RL},fi}}\right)}^{2},\left(\frac{S_{inc,f_{i}}}{S_{inc,RL,fi}}\right)\right\}\\ \;+\overset{300\;\mathrm{GHz}}{\underset{i>2\;\mathrm{GHz}}{\sum }}\left(\frac{S_{\mathrm{inc},f_{i}}}{S_{\mathrm{inc},\mathrm{RL},f_{i}}}\right)\;\leq 1 \end{array}$$
where *f_i_* is the frequency range, $E_{inc,f_{i}}$ is the electric incident field level, $E_{inc,RL,f_{i}}$ is the electric incident field reference level at $f_{i}$. $H_{inc,f_{i}}$ and $H_{inc,RL,f_{i}}$ are the magnetic field and reference levels at $f_{i}$, respectively. $S_{inc,f_{i}}$ and $S_{inc,RL,f_{i}}$ are the power density and its reference level at $f_{i}$, respectively.

## 3. Compliance Distance Calculations

In [22], the International Electrotechnical Commission (IEC) specified in their standard IEC62232 that the most accurate compliance boundary is obtained as an iso-surface shape that can be enclosed in volumes with simpler shapes to define more conservative boundaries, such as a box shape, which is suitable for the sector coverage antenna, with horizontal, side, and vertical directions. In this work, the box-shaped compliance boundary was used to take into account the horizontal and vertical beamwidths and the gains of the antennae. Following some related works [23,24,25,26,27], we used the compliance distance (R_CD_) to refer to the distance from the antenna at which TER = 1, which means that the accumulated exposure reaches the maximum limit. The TER was calculated on the basis of the power density S_inc_; therefore, Equation (2) was modified to Equation (3), as shown below [6].
(2)$$TER=\overset{300\;\mathrm{GHz}}{\underset{f>30\;\mathrm{MHz}}{\sum }}\left(\frac{S_{\mathrm{inc},f}}{S_{\mathrm{inc},\mathrm{RL},f}}\right)\;$$

At distance R (m) from the antennae, the S_inc_ is the power density, which can be expressed as:(3)$$S_{\mathrm{inc},f}=\frac{{\left(P_{T\;}.G_{A}\right)}_{f}}{4.\pi .R^{2}}$$

Equation (3) can be substituted in the nominator of Equation (2), so the TER would be:(4)$$TER=\overset{300\;\mathrm{GHz}}{\underset{f\;>30\;\mathrm{MHz}}{\sum }}\left(\frac{{\left(P_{T}.G_{A}\right)}_{f}}{4.\pi .R^{2}.S_{\mathrm{inc},\mathrm{RL},f}}\right)\;$$

At compliance distance R_CD_, the TER = 1, then:(5)$$R_{CD}={\left(\frac{1}{4\pi }\;\overset{300\;\mathrm{GHz}}{\underset{f\;>30\;\mathrm{MHz}}{\sum }}\frac{P_{T,f}.G_{A}{,}_{f}}{S_{\mathrm{inc},\mathrm{RL},f}}\;\right)}^{1/2}$$

With the introduction of 5G NR, which uses a high grade of massive Multi-Input Multi-Output (mMIMO), some recent studies have investigated EMF exposure by considering more factors [28,29,30,31,32], such as the actual radiated power, system utilization load, and spatial/time duty cycle [32,33]. Furthermore, according to IEC 62232 [22], EMF assessments for human exposure can be performed on the basis of actual situations. In [34,35], the authors investigated the time-averaged instantaneous radiation and the theoretical maximum exposure from 5G NR base stations, and their results stated that the actual maximum exposure was very low compared with the theoretical maximum exposure for NR mMMO. In this study, we used the power rating factor as a reduction in the total transmitted power to compute the compliance distance. Thus, Equation (5) becomes Equation (6):(6)$$R_{CD}={\left(\;\frac{1}{4\pi }\;\overset{300\;GHz}{\underset{f>30\;MHz}{\sum }}\frac{\rho_{r,f}.P_{T,f}.G_{A}{,}_{f}}{S_{inc,RL,f}}\;\right)}^{1/2}$$

## 4. Results and Discussion

A modern mobile wireless network is comprised of a combination of different site types and different access technologies that work together as a heterogeneous network. The RF planning teams in the MNOs select the appropriate site type according to many inputs, such as the target coverage range, number of users, forecasted traffic growth, total cost, and nature of the area (rural, urban, suburban, dense, indoor buildings, etc.). This study looked into multi-operator, multi-technology CDs for two types of sharing sites, macro and indoor IBS sites, which are currently the most commonly used.

### 4.1. Shared Macro Site

The macro site is a general solution that is used by all operators to provide coverage in populated areas [36]. The macro site has radio equipment with higher Tx power than other types of sites (micro, femto, and indoor), and the coverage range can reach long distances depending on the antenna height and site configurations. The macro site can be installed in different civil models, such as on monopoles, self-based towers, cells on wheels (COWs), wall-mounted antennae, or towers at rooftop sites. This study considered passive sharing, as shown in Figure 3 and illustrated in Figure 4.

Typical configurations of macro sites equipped with multi-technology solutions were used as input data for the calculations. The site was operated with six technologies: 2G GSM at 900 MHz (G900), 3G UMTS at 900 MHz (U900), 4G LTE at 800 MHz (L800), 4G LTE at 1800 MHz (L1800), 4G LTE at 2100 MHz (L2100), and 5G NR at 3500 MHz (N3500). The input data for each technology, including Tx/Rx, antenna gain, frequency bandwidth, system load, and transmitting power, are listed in Table 2. For N3500, which has a high grade of mMIMO (64T/64R), we used 0.22 as a realistic power factor. This was taken from the interesting results and findings of the studies [28,29,30]; it indicates that the realistic transmitted power of mMIMO reaches 16–22% of the maximum equipment power when it operates at a full system load.

The site configuration data were utilized in Equation (6) to compute the vertical and horizontal R_CD_ (R_DC-H_ and R_DC-V_), as displayed in Figure 5. The calculation outcome expressed that all horizontal R_DC-H_ distances were longer than the vertical R_DC-V_ distances, and the ICNIRP has a little more restricted R_CD_ compared with the FCC. In the main antenna horizontal direction, the general public ICNIRP R_CD-H_ was 14.81 m when there was a single MNO, while it increased to 29.6 m when 4 MNOs shared the macro site. The general public FCC R_CD-H_ was 13.8 m when there was a single MNO, and it increased to 27.7 m for 4 MNOs. Figure 5 summarizes the R_CD_ in both the vertical and horizontal directions for the GP.

Further processing was carried out to analyze the total emitted power density and related contribution to the total exposure ratio. Figure 6 depicts the power density PD (W/m^2^) calculated at distance (m) from the antennae in the presence of four MNOs, and the analysis showed that 5G N3500 had the highest PD among the technologies, and all technologies reached high levels at distances very close to the macro site’s antenna (less than 3 m).

With reference to the ICNIP for the GP, Figure 7 displays the exposure ratio for each technology calculated by Equation (4) in the horizontal direction and TER for the macro site shared by a single MNO, two MNOs, three MNOs, and four MNOs.

The results showed that 5G N3500 had the highest contribution at the R_CD_ with 34%, followed by 4G L800 with 24%, 4G L1800 and L2100 with 9–10% each, and 2G G900 and 3G U900 with 11% each. Similarly, at the R_CD_ based on FCC limits, the highest contributor was N3500 (39%), followed by L800 (21%), L1800 and L2100 (11% each), and G900 and U900 (9% each). Figure 8 summarizes the exposure contributions of each technology at the R_CD_, and these were in line with the results found in [36,37] for the power density.

Equation (6) gives shorter compliance distances compared with the calculations performed using Equation (4) because it is based on realistic rated power for the mMIMO rather than the conservative power levels used in Equation (4). The R_CD_ increased with the increase in the number of MNOs sharing the same location; Table 3 summarizes the percentage of the distance increase compared with a single MNO with reference to the ICNIRP and FCC limits. It shows a +41% increase for 2 MNOs for all standard limits, +73% for 3 MNOs, and +100 for 4 MNOs.

The results in Table 4 can be scaled up to any number N of MNO operators sharing the same site, assuming all operators use the same configuration and technologies. In this case, Equation (6) and can be written for one MNO as:(7)$$CD_{1}={\left(\;\frac{1}{4\pi }\;\overset{300\;GHz}{\underset{f>30\;MHz}{\sum }}\frac{\rho_{r,f}.a_{b,f}.P_{T,f}.G_{A}{,}_{f}}{S_{inc,RL,f}}\;\right)}^{1/2}$$

For N number of MNOs, Equation (7) becomes Equation (8):(8)$$CD_{N}={\left(\;\frac{1}{4\pi }\;\overset{300\;GHz}{\underset{f>30\;MHz}{\sum }}N.\frac{\rho_{r,f}.a_{b,f}.P_{T,f}.G_{A}{,}_{f}}{S_{inc,RL,f}}\;\right)}^{1/2}$$
(9)$$CD_{N}=N^{1/2}.CD_{1}$$

Equation (9) represents a simplified formula that gives the compliance distance based on the EMF Site-Sharing Scale (EMF-SSS formula). In reality, it is very hard to find a group of MNO operators that use the same configuration for multi-technology, including the power rating and system load. However, Equation (9) is useful for predicting the CD and indicating the expected increase in the CD when accommodating more than one MNO at the same site, and Figure 9 plots this relationship using Table 2 as input data for Equation (8).

### 4.2. Shared Indoor IBS Site

Most of the traffic in mobile networks is generated from indoor houses and buildings [38], and in some dense areas, huge buildings suffer from weak indoor single strength due to construction loss [39], especially in public places such as airports, commercial malls, cinemas, hospitals, hotels, universities, and commercial buildings. The in-building solution (or indoor-based solution) provides indoor coverage by distributing the signal to several antennae via a series of hubs (splitters), as shown in Figure 10. The IBS gives additional leverage to strengthen the level and quality of wireless signals, thereby ensuring smooth wireless communication for mobile network users. The operators install the IBS inside massive buildings and high-rise towers, where the received coverage from outdoor macro sites is weak due to the construction penetration losses. The IBS combines all the technologies into one Distributed Antenna System (DAS) in which each antenna provides coverage for a certain zone as a hotspot [17,40].

The IBS-DAS consists of many connected antennae that are shared and energized from the same radio transmitters in the splitting system. In this study, we looked at radiation from a single antenna (one zone) using high-density configurations of a single IBS site outfitted with multi-technology. The IBS was operated with six technologies: G900, U900, L800, L1800, L2100, and N3500. The input data for each technology are listed in Table 4. They are similar to those of the macro site but have less Tx power, lower antenna gains, and no mMIMO in 5G.

**Table 4 sensors-23-01588-t004:** Configuration setup for shared indoor IBS.

Site Setting	2G G900	3G U900	4G L800	4G L1800	4G L2100	5G N3600
Freq. Band (MHz)	900	900	800	1800	2100	3500
Freq. BW (MHz)	5	5	10	20	20	100
Total Tx	2	1	2	2	2	4
Total Rx	2	1	2	2	2	4
Power Tx	1.3 W	1.3 W	1.3 W	1.3 W	1.3 W	2 W
System Load	95%	95%	95%	95%	95%	95%
Ant. Gain	8 dBi	8 dBi	8 dBi	10 dBi	10 dBi	11.1 dBi

Similar to the macro results, the IBS results expressed that all R_DC-H_ distances were longer than the vertical R_DC-V_ distances, and the ICNIRP had a slightly more restricted R_CD_ compared with the FCC. The general public ICNIRP R_CD-H_ was 0.89 m when there was a single MNO, but it increased to 1.74 m for four MNOs. Additionally, for the general public, R_CD-H_ was 0.4 m when there was a single MNO, while it increased to 0.79 m for four MNOs. Figure 11 shows the R_CD_ for the general public in both the vertical and the horizontal direction.

In addition, further processing was carried out to analyze the power density and its related contribution to the total exposure ratio. Figure 12 presents the power density PD (W/m^2^) calculated at a distance (m) from the antennae, and the analysis showed that 5G N3500 had the highest PD compared with the other technologies; it reached high levels at distances very close to the DAS antenna.

With reference to the ICNIP for the GP, Figure 13 displays the exposure ratio for each technology calculated using Equation (4) in the horizontal direction and the total TER for the IBS site shared by a single MNO, two MNOs, three MNOs, and four MNOs.

The results showed that 5G N3500 MHz had the highest contribution to the TER at the compliance distances for the ICNIRP with a contribution of 26%, followed by 4G L800 with 20%, 2G G900 and 3G U900 with 18% each, 4G L1800 with 9%, and 4G L2100 with 8%. Similarly, the N3500 had the highest contribution (30%), followed by 4G L800 (18%), 2G G900 and 3G U900 (16% each), and 4G L1800 and 4G L2100 (both with 10%). Figure 14 summarizes the exposure contributions of each technology at the R_CD_, and these were the expected results because the IBS site does not use a high grade of mMIMO.

Similar to that of the shared macro site, with an increase in the number of MNOs, the R_CD_ of the IBS increased with the same percentage for all references. Table 5 summarizes the percentage of the distance increase compared with a single MNO with reference to the ICNIRP and FCC limits. It shows an increase of +41% for two MNOs, +73% for three MNOs, and 100% for four MNOs.

Additionally, in practice, it is difficult to find groups of MNO operators that share IBS sites with the same configurations. However, using Table 4 as the input, Equation (9) was used to scale up the increase in R_CD_ with the increased N of MNOs sharing the IBS, as shown in Figure 15.

## 5. Conclusions

The EMF exposure compliance boundary for multi-technology, multi-operator site sharing was discussed and analyzed using calculations that considered realistic configurations, and the study covered two types of site sharing: macro and indoor IBS sites. For a typical site operating with 2G/3G/4G/5G, the compliance distance results for the general public zone were less than those for the occupational zone. The ICNIRP limits were slightly more restricted than the FCC limits. For a macro site with a single MNO, the general public’s horizontal R_CD_ distances were 14.8 m and 13.8 m according to the ICNIRP and FCC, respectively. Furthermore, for a macro site shared by four MNOs, the R_CD_ distances were 29.61 m and 27.69 m according to the ICNIRP and FCC, respectively. In addition, for all shared macro site scenarios, the 5G N3500 MHz had the highest contributions to the TER of 34% and 39% at the horizontal R_CD_ according to the ICNIRP and the FCC, respectively. For the indoor IBS site with a single MNO, the general public’s horizontal R_CD_ distances were 0.87 m and 0.8 m according to the ICNIRP and FCC, respectively. In addition, for an IBS shared by four MNOs, the R_CD_ distances were 1.74 m and 1.6 m according to the ICNIRP and FCC, respectively. Moreover, for all shared IBS site scenarios, the 5G N3600 MHz had the highest contributions to the TER of 26% and 30% at the horizontal compliance distances according to the ICNIRP and FCC, respectively. In the future, the research team will investigate the R_CD_ for the accumulated radiation considering the neighboring sites. 

## Figures and Tables

**Figure 1 sensors-23-01588-f001:**
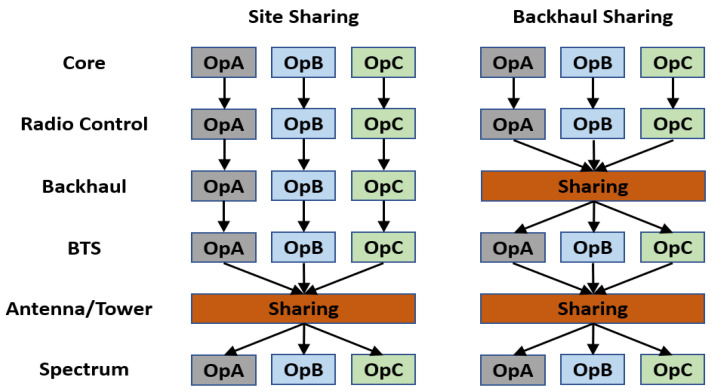
Illustration of passive sharing between three operators.

**Figure 2 sensors-23-01588-f002:**
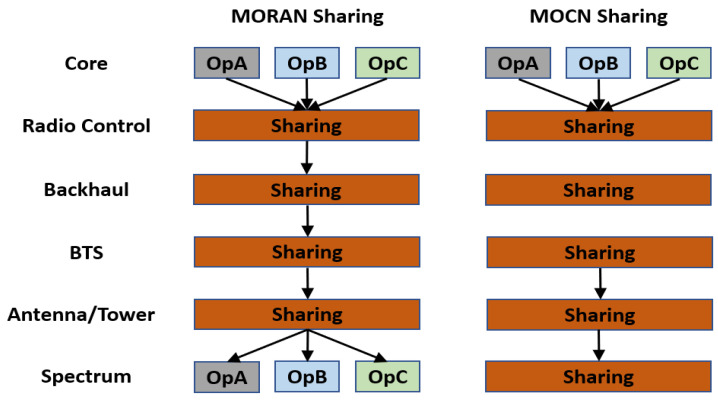
MORAN and MOCN active sharing between three operators.

**Figure 3 sensors-23-01588-f003:**
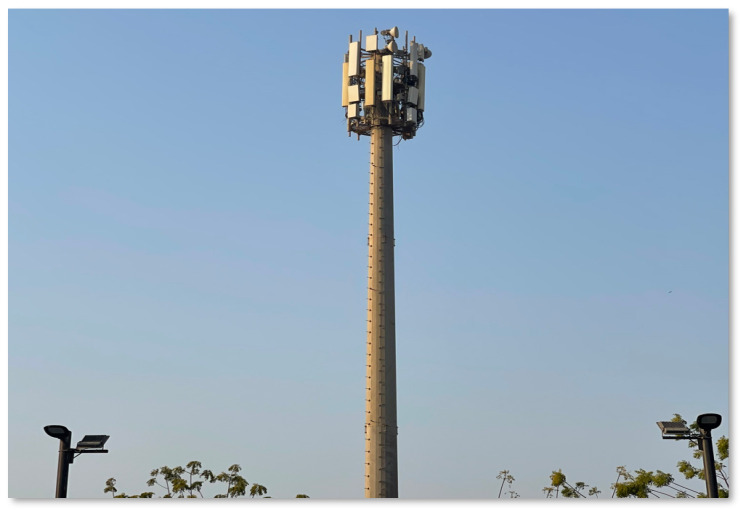
Photograph of macro site shared by three MNOs.

**Figure 4 sensors-23-01588-f004:**
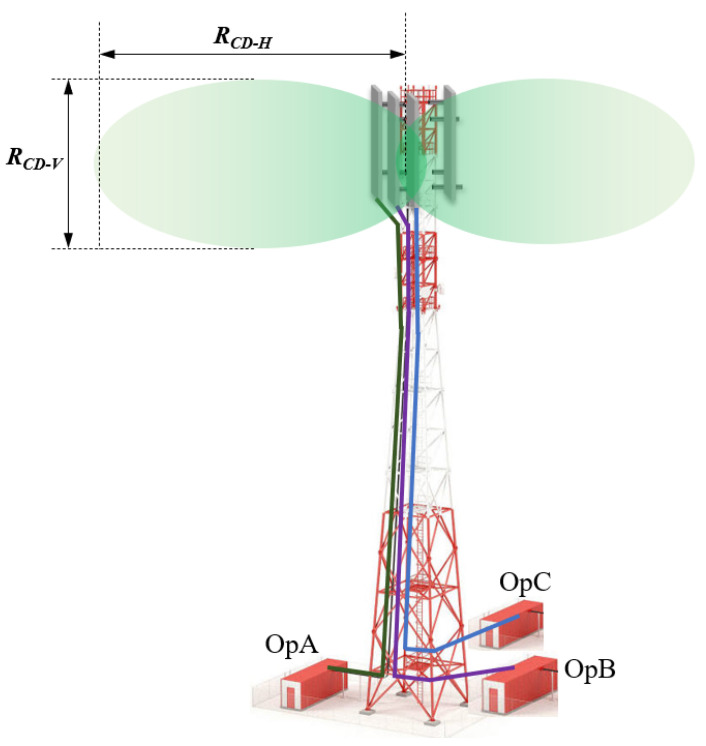
The horizontal R_CD-H_ and vertical R_CD-V_ compliance distances for macro site installed on rooftop tower.

**Figure 5 sensors-23-01588-f005:**
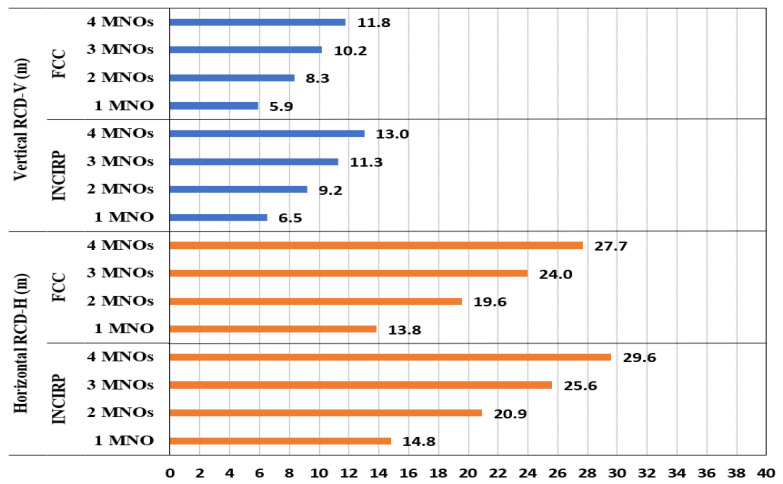
The R_CD_ results for multi-technology multi-operator sharing a macro site, referencing the ICNIRP and FCC standards limits for GP.

**Figure 6 sensors-23-01588-f006:**
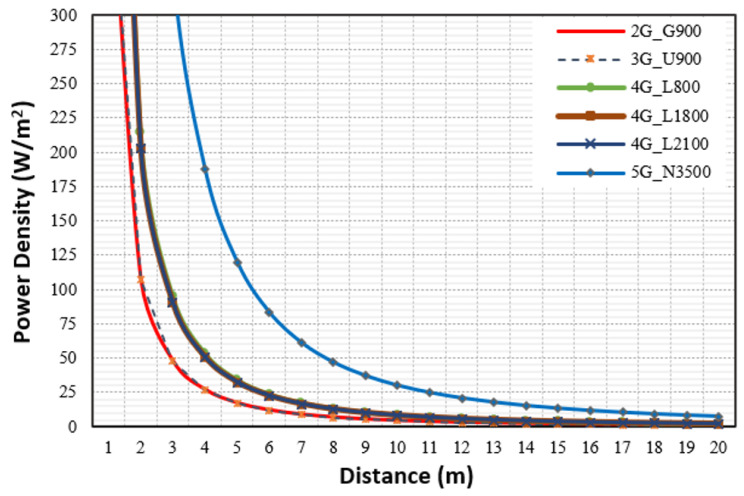
The power densities (W/m^2^) in the main direction for the macro site shared by 4 MNOs.

**Figure 7 sensors-23-01588-f007:**
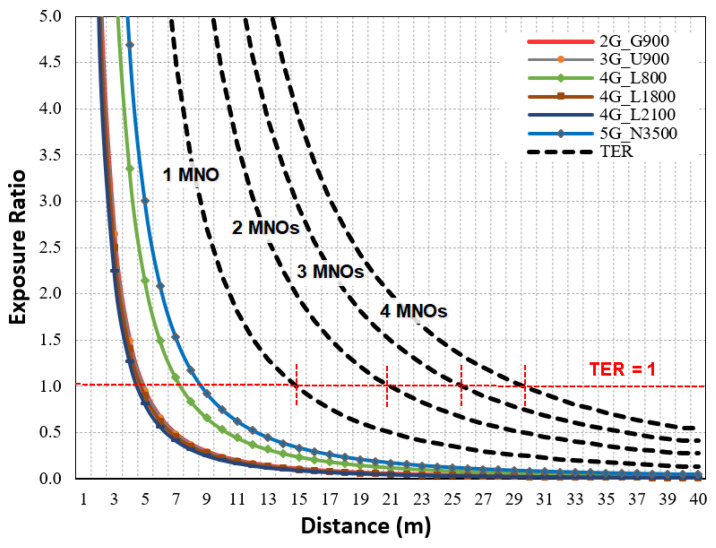
The exposure ratio in the main direction for multi-technology macro site with reference to ICNIRP limits for GP.

**Figure 8 sensors-23-01588-f008:**
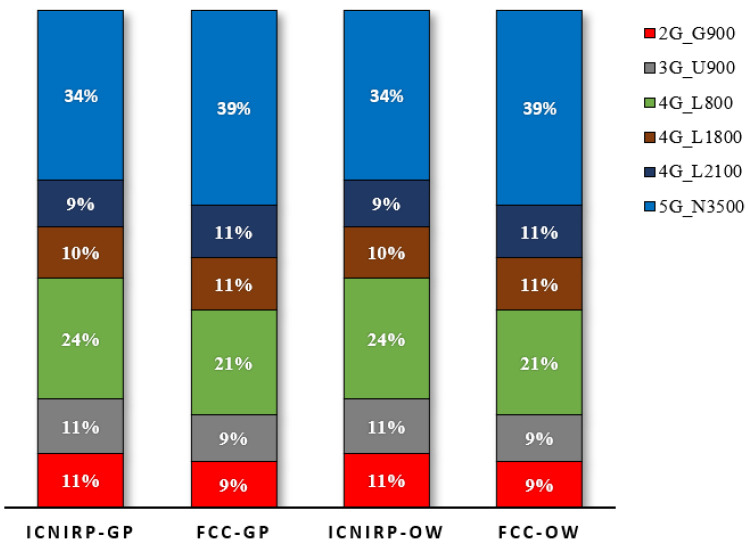
The EMF radiation contribution at the R_CD_ in the horizontal direction for the macro site shared by 4 MNOs.

**Figure 9 sensors-23-01588-f009:**
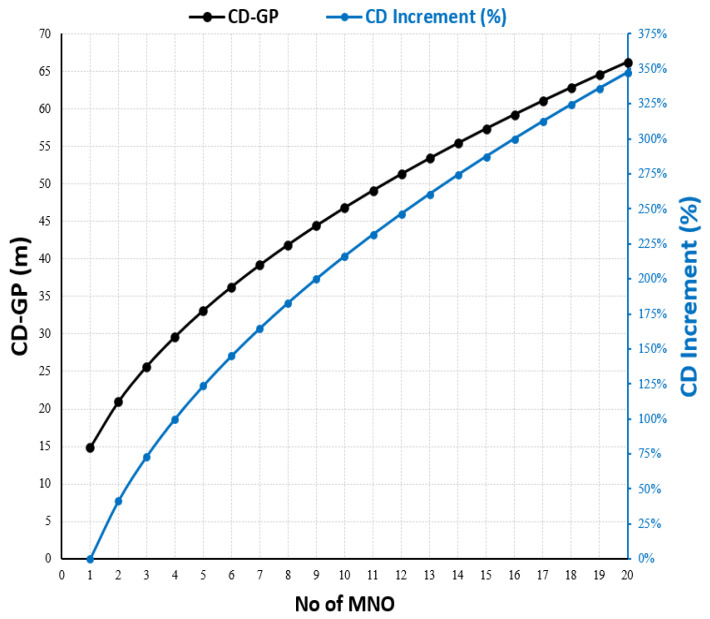
The compliance Distance for GP at macro site shared by N number of MNOs using the typical RF configuration in Table 4.

**Figure 10 sensors-23-01588-f010:**
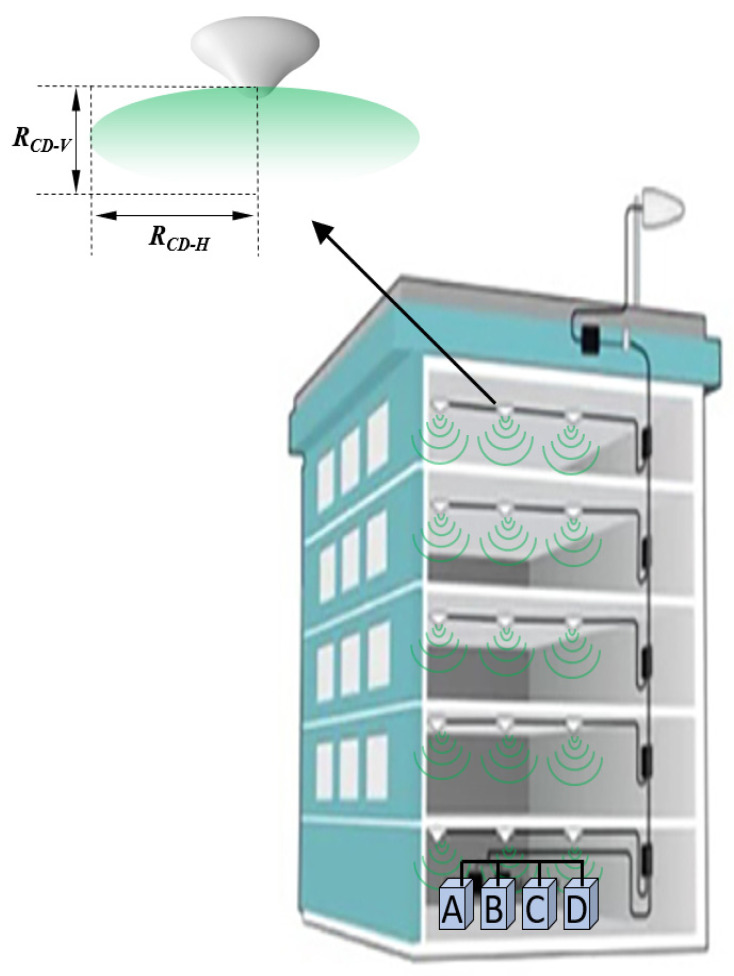
The horizontal R_CD-H_ and vertical R_CD-V_ compliance distances for one IBS-DAS antenna installed on the ceiling of a building floor for IBS shared by 4 operators (A, B, C, and D).

**Figure 11 sensors-23-01588-f011:**
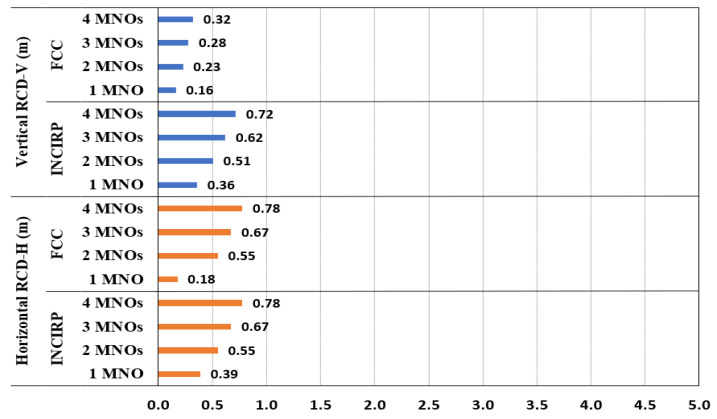
The R_CD_ for shared IBS-DAS site, referencing the ICNIRP and FCC limits for GP.

**Figure 12 sensors-23-01588-f012:**
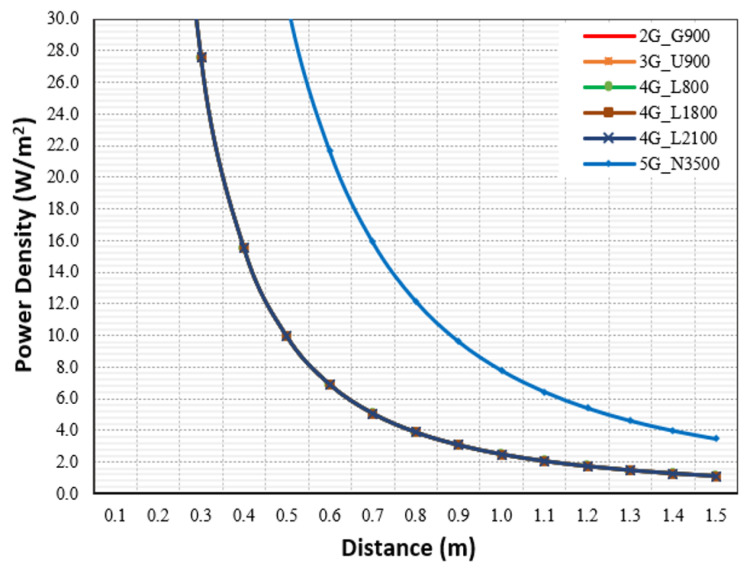
The PD (W/m^2^) from the antenna for the shared IBS-DAS site.

**Figure 13 sensors-23-01588-f013:**
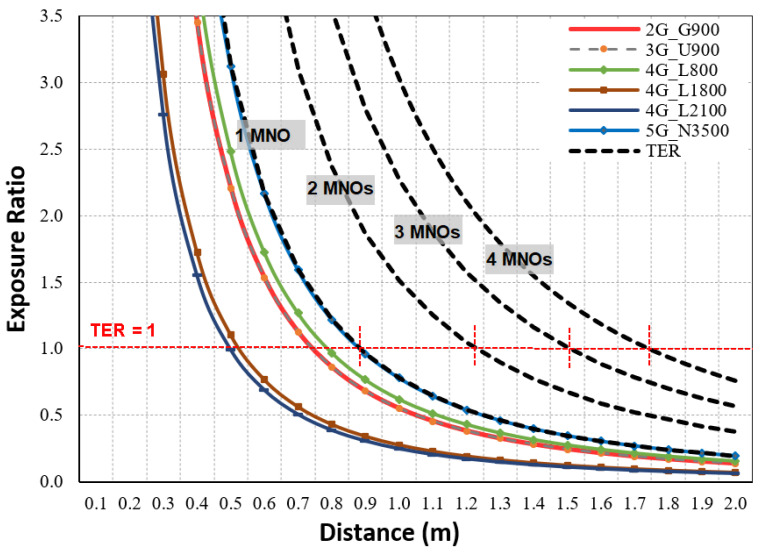
The exposure ratio at the main direction for multi-technology IBS site, referencing ICNIRP limits for GP.

**Figure 14 sensors-23-01588-f014:**
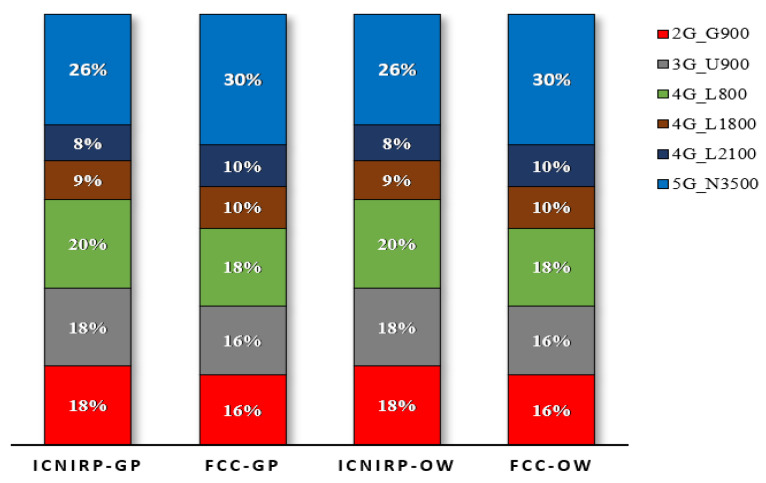
The EMF radiation contribution from each technology at CD for shared IBS site, referencing ICNIRP and FCC limits for GP and OW.

**Figure 15 sensors-23-01588-f015:**
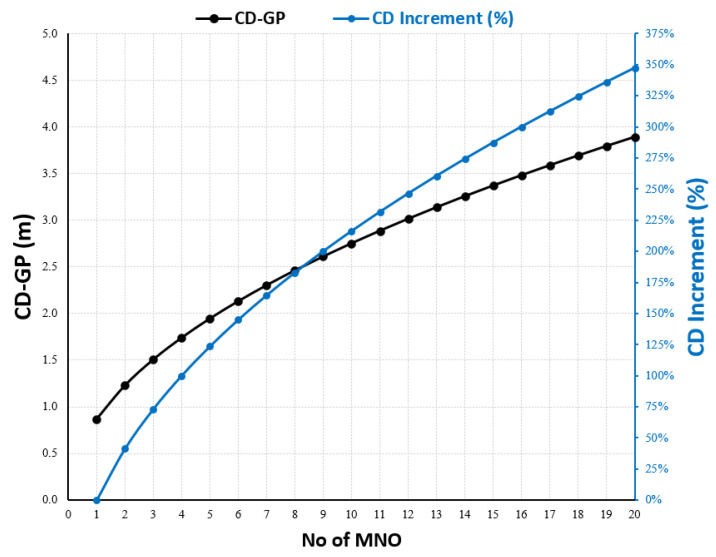
The R_CD_ increment at IBS site shared by N number of MNOs using the typical RF configuration in Table 4.

**Table 1 sensors-23-01588-t001:** The ICNIRP and FCC maximum limits for frequencies 0.1 MHz to 300 GHz.

Exposure Limit	Freq. Range	E-Field (V/m)	H-Field (A/m)	PD (W/m^2^)
**ICNIRP-OW**	0.1–30 MHz	$660/f_{M}{}^{0.7}$	$4.9/f_{M}$	*NA*
	300–400 MHz	*61*	*0.16*	*10*
	400–200 MHz	$3f_{M}{}^{0.5}$	$0.008f_{M}{}^{0.5}$	$f_{M}/40$
	2–300 GHz	*NA*	*NA*	*50*
**ICNIRP-GP**	0.1–30 MHz	$300/f_{M}{}^{0.7}$	$2.2/f_{M}$	*NA*
	300–400 MHz	*27.7*	*0.073*	*2*
	400–200 MHz	$1.375f_{M}{}^{0.5}$	$0.0037f_{M}{}^{0.5}$	$f_{M}/200$
	2–300 GHz	*NA*	*NA*	*10*
**FCC-OW**	0.3–0.3 MHz	*614.0*	*1.630*	*100.0*
	3.0–30 MHz	*1842/f*	*4.89/f*	*900/* $f^{2}$
	30–300 MHz	*61.40*	*0.163*	*1.00*
	0.3–1.5 GHZ	*-*	*-*	*f/300*
	1.5–100 GHz	*-*	*-*	*5.00*
**FCC-GP**	0.3–1.34 MHz	*614.0*	*1.630*	*100*
	1.34–30 MHz	*824/f*	*2.19/f*	*180/* $f^{2}$
	30–300 MHz	*27.50*	*0.0730*	*0.20*
	0.3–1.5 GHz	*-*	*-*	*f/1500*
	1.5–100 GHz	*-*	*-*	*1.00*

**Table 2 sensors-23-01588-t002:** Configuration setup for macro site.

Site Setting	2G G900	3G U900	4G L800	4G L1800	4G L2100	5G N3500
Freq. Band (MHz)	900	900	800	1800	2100	3500
Freq. BW (MHz)	5	5	10	20	20	100
Total Tx	2	1	2	2	4	64
Total Rx	2	1	2	2	4	64
Power Tx	40 W	40 W	80 W	80 W	80 W	160 W
System Load	95%	95%	95%	95%	95%	95%
Ant. Gain	17 dBi	17 dBi	16.7 dBi	16.6 dBi	17 dBi	24.8 dBi

**Table 3 sensors-23-01588-t003:** The R_CD_ for macro site shared by up to 4 MNOs, referencing the ICNIRP and FCC limits.

Limits	Compliance Distance (m)						
	1MNO	2xMNO	%	3xMNO	%	4xMNO	%
**INCIRP**	14.8	20.9	+41%	25.6	+73%	29.6	+100%
**FCC**	13.8	19.6	+41%	24.0	+73%	27.7	+100%
**INCIRP**	6.6	9.4	+41%	11.5	+73%	13.2	+100%
**FCC**	6.2	8.8	+41%	10.7	+73%	12.4	+100%
**INCIRP**	6.5	9.2	+41%	11.3	+73%	13.0	+100%
**FCC**	5.9	8.3	+41%	10.2	+73%	11.8	+100%
**INCIRP**	2.9	4.1	+41%	5.0	+73%	5.8	+100%
**FCC**	2.6	3.7	41%	4.6	73%	5.3	100%

**Table 5 sensors-23-01588-t005:** DC for IBS site shared by up to 4 MNOs, referencing the ICNIRP and FCC limits.

Limits	Compliance Distance (m)						
	1MNO	2xMNO	%	3xMNO	%	4xMNO	%
**INCIRP**	0.9	1.2	+41%	1.5	+73%	1.7	+100%
**FCC**	0.8	1.1	+41%	1.4	+73%	1.6	+100%
**INCIRP**	0.4	0.6	+41%	0.7	+73%	0.8	+100%
**FCC**	0.4	0.5	+41%	0.6	+73%	0.7	+100%
**INCIRP**	0.4	0.6	+41%	0.7	+73%	0.8	+100%
**FCC**	0.4	0.5	+41%	0.6	+73%	0.7	+100%
**INCIRP**	0.2	0.3	+41%	0.3	+73%	0.4	+100%
**FCC**	0.2	0.2	+41%	0.3	+73%	0.3	+100%

## Data Availability

Not applicable.

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
