# Peer review of "Multi-Technology Multi-Operator Site Sharing: Compliance Distance Analysis for EMF Exposure"

_sensors, 2023, doi:10.3390/s23031588_

Round 1
Reviewer 1 Report
The paper is original and well organized with mathematical analysis and quality figures representing the outcomes. But still there is one thing missing ie. a performance comparison table/
It is advised authors to add a table of comparison of performance parameters between proposed work and existing works in the open literature.
Authors to look at the references there are typos Ex: Check Reference-1: Author1
Reviewer 2 Report
This article presents a novel system for Multi-Technology Multi-Operator Site Sharing. This system is relatively simple and accurate and its operating principles are well demonstrated. Its use may be of interest to a great number of applications among researchers studying compliance distance analysis for EMF exposure. From a technical point of view, this article is highly interesting for the real operating set-up. Besides, the obtained results are quite satisfactory. The manuscript is well-organized, and the subject is well-studied. However, I would like to make a few comments for the authors:
- if some of the equations used in this paper are not newly derived for this manuscript, the references for these equations should be given in the appropriate lines for clear understanding.
- How the author explains the obtained results in terms of performance metrics. What about the previous studies' achievements compared to the proposed one?
- The format of the references is changing. Please use the same format for all publications in the references.
- The authors have forgotten to cite the following paper, while they are talking about the site location for 5G:
"Performance analysis of the TOA cooperative localization estimation algorithm for 5G cellular networks, in IEEE 26th Signal Processing and Communications Applications Conference (SIU), pp. 1-4, May 2018"
- Even if the theoretical background is quite missing the real operating environment and applications on it are so attractive.
For the current version, the authors should update the manuscript with the help of the comments stated above.
Thanks for your consideration.
